# Predictors of Willingness to Receive COVID-19 Vaccine: Cross-Sectional Study of Palestinian Dental Students

**DOI:** 10.3390/vaccines9090954

**Published:** 2021-08-26

**Authors:** Elham Kateeb, Mayar Danadneh, Andrea Pokorná, Jitka Klugarová, Huthaifa Abdulqader, Miloslav Klugar, Abanoub Riad

**Affiliations:** 1Oral Health Research and Promotion Unit, Al-Quds University, Jerusalem 51000, Palestine; ekateeb@staff.alquds.edu (E.K.); mayar.danadneh@students.alquds.edu (M.D.); 2Public Health Committee, World Dental Federation (FDI), 1216 Geneva, Switzerland; 3Public Policy Center, University of Iowa, Iowa, IA 52242, USA; 4Czech National Centre for Evidence-Based Healthcare and Knowledge Translation (Cochrane Czech Republic, Czech EBHC: JBI Centre of Excellence, Masaryk University GRADE Centre), Institute of Biostatistics and Analyses, Faculty of Medicine, Masaryk University, 62500 Brno, Czech Republic; apokorna@med.muni.cz (A.P.); klugarova@med.muni.cz (J.K.); abanoub.riad@med.muni.cz (A.R.); 5Department of Nursing and Midwifery, Masaryk University, 62500 Brno, Czech Republic; 6Department of Public Health, Masaryk University, 62500 Brno, Czech Republic; 7International Association of Dental Students (IADS), 1216 Geneva, Switzerland; vpsr@iads-web.org

**Keywords:** COVID-19 vaccines, cross-sectional studies, decision making, education, dental, students, dental, mass vaccination, social determinants of health

## Abstract

The overarching aim of this study was to assess the predictors related to the willingness of Palestinian dental students to receive the COVID-19 vaccine when it becomes available. A cross-sectional study was conducted among a universal sample of dental students in the Palestinian territories. Willingness to get the COVID-19 vaccine was related to the following factors: Demographic characteristics, COVID-19-related experiences, beliefs and knowledge about the vaccine, attitudes toward vaccinations in general, and other factors outlined by the WHO SAGE Vaccination Hesitancy Questionnaire. Four hundred and seventeen students completed the questionnaire (response rate = 41.7%). In general, 57.8% (*n* = 241) were willing to take the COVID-19 vaccine when it became available to them, 27% (*n* = 114) were hesitant, and 14.9% (*n* = 62) were not willing to get vaccinated. The final regression model explained 46% of the variation in the willingness to receive the COVID-19 vaccine as follows: Attitudes towards new vaccines (β = 6.23, *p* < 0.001), believing in a favorable risk–benefit ratio (β = 5.64, *p* < 0.001), trust in the pharmaceutical industry (β = 5.92, *p* = 0.001), believing that natural immunity is better than being vaccinated (β = −4.24, *p* < 0.001), and having enough information about the vaccine (β = 4.12, *p* < 0.001). Adequate information about vaccines, their risk–benefit ratios, and natural and acquired immunity are important to build trust and favorable attitudes towards vaccines among future dentists.

## 1. Introduction

The novel severe acute respiratory syndrome coronavirus 2 (SARS-CoV-2) causing coronavirus disease 2019 (COVID-19) emerged in December 2019 and was declared as a pandemic in March 2020 [1]. As of 2 July 2021, 182 million cumulative cases and almost four million deaths have been recorded worldwide [2]. Global collaborative efforts had led to the rapid development of vaccines against COVID-19.

In December 2020, several vaccines were authorized worldwide and approved by the World Health Organization (WHO) to prevent COVID-19 infection [3]. Vaccination campaigns have begun in various countries at different speeds using different implementation strategies depending on availability, rollout speed, and acceptance rates among people [4].

The data available at the time of writing this manuscript indicate that 343,710 confirmed cases and 3845 deaths due to COVID-19 occurred in the Palestinian territories [2]. The formal vaccination campaign started in the second week of March 2021, targeting healthcare workers primarily, then the elderly (over 70 years old) and patients with chronic disease [5]. Vaccines became available in Palestine through the international vaccine-sharing scheme (COVAX), other countries’ donations, and direct purchasing from pharmaceutical companies. According to the Palestinian Ministry of Health (MoH) press releases, as of 8 July 2021, 536,130 took the Covid-19 vaccine, with 385,465 of them receiving two doses; thus, almost 10% of the Palestinians in the West Bank and Gaza Strip received at least one dose of the vaccine. There have been four types of vaccine available since March 2021, Pfizer-BioNTech, AstraZeneca-Oxford, Sputnik V, and Sinopharm, administered through assigned MoH public clinics and following a two-dose schedule to achieve the highest effectiveness [5].

When issues related to availability, short supply, and other vaccination rollout logistics such as the safe and secure transporting and delivering the vaccine and ensuring adequately trained manpower for vaccine administration are solved, the main barrier to delivering the vaccine adequately will be reaching those who are reluctant to become vaccinated [6,7]. Vaccine hesitancy (delay in acceptance or refusal of vaccination despite the availability of vaccination services) has been a concern since even before the current pandemic [8]. The WHO declared that this uncertainty is among the top ten global health threats since 2019 [9].

Studies from different parts of the world found that willingness for vaccination varies widely depending on the vaccines’ effectiveness and safety profile [10,11,12,13]. A recent global review found that 72% of people would take a vaccine against COVID-19 if it were proven safe and effective, but willingness varied widely between the included nations [14]. Differences in acceptance rates among the 19 countries included in this survey ranged from almost 90% (in China) to less than 55% (in Russia) [14].

In general, factors related to vaccine hesitancy, as reported in the literature, include religious reasons, personal beliefs, and safety concerns due to widespread myths, including the association of vaccines with autism, brain damage, and other conditions [15]. Mistrust towards healthcare professionals and health authorities and governments in general were documented in the literature as major influencers in intentions to get vaccinated [16,17].

Additionally, vaccine hesitancy level differs across the different vaccines; the factors related to acceptance of the influenza vaccine, for example, may not apply to the new COVID-19 vaccines [18]. Therefore, investigating the factors related to the COVID-19 vaccine in specific contexts and cultures is necessary to identify factors influencing the decision to become vaccinated or not.

Healthcare professionals (HP) are expected to have high levels of vaccine acceptance due to the nature of their work and the knowledge they have about the science behind vaccines and their effectiveness. However, the willingness of HP worldwide to get the vaccine has been unclear and has varied over time and among different contexts [19,20,21,22,23].

Dentistry is one of the health professions most affected by the current pandemic. Dental professionals, including dental students in clinical years, usually work in close proximity to patients using procedures that expose them to high levels of aerosols, droplets, and oral fluids. This may cause additional risks of viral exposure and transmission from infected patients to the dental team, and vice versa, and subsequently to other patients, if appropriate infection control measures are not undertaken [24,25,26,27]. The previous reports indicated that dentistry was the most at-risk profession for SARS-CoV-2 compared to other various occupations [24,25,26,27].

The initial response of dentists and dental students in Palestine to the current pandemic showed high risk perception of COVID-19 and reflectance to treat patients due to fear of transmitting the virus to family and friends [26,27,28]. A recent study conducted among Palestinian dental students demonstrated high prevalence of psychological distress during the period of the lockdown related to the COVID-19 pandemic [29].

In addition to the increased occupational risks of the dental practice, COVID-19 brought a new challenge for oral health professionals through its oral symptoms that still have no clear pathophysiology or prevalence. COVID-19-associated oral manifestations were increasingly reported in the last months, e.g., loss of taste (dysgeusia), perioral and intraoral ulcers, oral candidiasis, oral mucositis, and parotid gland inflammation; therefore, they are widely viewed as a demanding knowledge gap that requires rigorous investigation by dental researchers and practitioners [30,31,32,33,34,35,36].

Reports regarding the willingness of dentists and dental students to become vaccinated varied across different countries, according to the economic status of the country surveyed, and among individuals, based on the inadequacy of knowledge about vaccines, and the mistrust of governments and the pharmaceutical industry [37,38].

Most studies that assessed vaccine hesitancy used the proxy “willingness to get the vaccine when it becomes available”. Although willingness may not always correlate with actual behavior, including for vaccination, it is still a good indicator of acceptance and can give public health campaigns advance notice of whom to target in their vaccination promotion programs [39,40].

Therefore, this study aimed to assess the predictors related to willingness to receive the COVID-19 vaccine when it becomes available among a universal sample of dental students in different educational institutions in the West Bank and Gaza areas of the Palestinian territories. This study specifically assessed factors related to students’ knowledge, beliefs, and attitude and their association with the decision to get vaccinated. At the time of data collection in February 2021, a formal vaccination campaign had not started yet in Palestine. We hypothesized that factors such as trust, beliefs, and level of knowledge are influential in dental students’ decisions to become vaccinated.

## 2. Materials and Methods

### 2.1. Design

Data for the current study were extracted from a global cross-sectional multicounty survey conducted in the months of February and March of 2021 by the International Association of Dental Students (IADS) [41]. The global study was coordinated by the IADS national scientific committees in 22 countries and aimed to evaluate the dental students’ hesitancy levels towards COVID-19 vaccines. An online self-administered questionnaire (SAQ) that included closed-ended multiple-choice items was developed through KoboToolbox (Harvard Humanitarian Initiative. Cambridge, MA, USA, 2021) [42].

### 2.2. Participants

The questionnaire was sent out to undergraduate dental students and interns in the four Colleges of Dentistry in the Palestinian territories: Al-Quds University, the Arab American University in the West Bank area, Al-Azhar University, and the University of Palestine in Gaza Strip. The students who were enrolled in the 4 dental colleges in the academic year 2020/2021 and attended any year of the five academic levels of the dental surgery degree in Palestine or attended the internship year were eligible to participate in this study. The “clinical internship” is an obligatory year that students need to complete immediately after finishing their formal training before getting their full licensure to practice (Figure 1).

The official students’ Facebook (FB) groups for each college were used as our sampling frames, and two reminder messages, one week apart, were sent through the messenger application. The total number of students and interns in the Palestinian territories is 3650; 96% of them (*n* = 3500) could be found in these FB groups. A minimum sample of 379 was calculated using the Epi Info™ Version 7.2.4.0 online calculator to achieve a 95% confidence level and a 5% margin of error [43].

### 2.3. Instrument

The SAQ included 20 multiple-choice questions that covered four sections: (1) Demographic data including gender, age, academic level, or professional status (as an intern or a fresh graduate); (2) COVID-19-related experience including the previous infection, providing care to a COVID-19 patient, having a COVID-19 patient within the students’ close social circle, and having a deceased COVID-19 patient within the students’ close social circle; (3) willingness to get the COVID-19 vaccine; and (4) factors influencing the decision to get the COVID-19 vaccine and students’ attitudes towards new vaccines in general.

An expert panel consisting of two professors and two senior researchers selected the questions for section 4 from a validated instrument published by the WHO-SAGE (Vaccine Hesitancy Survey Questions Related to SAGE Vaccine Hesitancy Matrix) [44]. Additional pilot testing was carried out to test face validity and item reliability for the whole instrument by 18 dental students who were invited to fill in the questionnaire twice with a minimum interval of 48 h. The mean Cohen’s kappa coefficient of the test–retest was 81.83 ± 0.16 (0.55—1.0), indicating very good reliability [38].

### 2.4. Ethical Considerations

All subjects completed digital informed consent that emphasized the voluntary nature of their participation and the measures that were taken to ensure their confidentiality and privacy prior to filling out the questionnaire. The study protocol had been reviewed and approved by the Ethics Committee of the Faculty of Medicine, Masaryk University (MUNI) on 20 January 2021, with reference No. 4/2021. Administrative approval was obtained from the Deanship of Scientific Research at Al-Quds University to collect data in Palestine. The questionnaire was anonymous, and the study data were collected and managed by MUNI in full compliance with the European General Data Protection Regulation 2016/679 (GDPR) [45].

### 2.5. Statistical Analysis

Statistical analysis was conducted using the Statistical Package for the Social Sciences (SPSS) version 27.0 (SPSS Inc., Chicago, IL, USA, 2020) [46]. Descriptive statistics, frequencies, percentages, cumulative percentages, means, and standard deviations were generated for all study variables, both independent and dependent variables [38]. Independent variables included all variables in sections 1, 2, and 4. For analysis purposes, students in the “Academic Level” variable were categorized into two levels: Preclinical (first year, second year, third year) and clinical (fourth year, fifth year, interns, and fresh graduates). This reflects when clinical courses are introduced for these students, which usually happens in the 4th year with minimal clinical training before that.

Descriptive statistics also described our main dependent variable, “Willingness to get the vaccine when it becomes available”, which was measured on a 5-point Likert scale (1 = Strongly Disagree, 2 = Disagree, 3 = Not Sure, 4 = Agree, 5 = Strongly Agree). For descriptive statistics purposes, the dependent variable in this study was further categorized into three levels: “willing to get the vaccine”, which included the two ratings: Strongly Agree and Agree, “Hesitant to get the vaccine”, which included “Not sure”, and “Unwilling to get the vaccine”, which included “Strongly Disagree and Disagree”.

Bivariate analysis was conducted using Spearman’s correlation, the Mann–Whitney (U) test, the Kruskal–Wallis (H) test, and the Chi-squared test to assess the association between our dependent variable (the 5-point rating format) and other independent variables. The significance level was set to (*p*) ≤ 0.05 [38].

Finally, β statistics from the multiple linear regression model of the dependent variable, “Willingness to get the COVID-19 vaccine”, were calculated. The dependent variable used here was a five-point scale treated as a continuous variable in the regression model. Predictor variables that were found statistically significant in the bi-variable analysis were entered into the regression model using a stepwise technique and confirmed by back-ward and forward regression analysis.

## 3. Results

Four hundred and seventeen students from the four colleges of Dentistry in the Palestinian territories, including the West Bank and Gaza, completed the questionnaire (response rate = 41.7%). Seventy-one percent (*n* = 295) of the sample was female students, which reflects the real proportion of female students in Palestinian dental schools (70% of total graduates are females). Almost 48% (*n* = 202) of the sample were in preclinical years and 52% (*n* = 215) in clinical years, including the internship year. While 22 years old was the median age of the global sample, 86.1% of the Palestinian students were aged 22 years or below (Table 1).

Nineteen percent of our sample (*n* = 81) were aware that they had been infected by the SARS-CoV-2, 90% (*n* = 375) knew someone who had been infected in their close circle, and 51% (*n* = 211) knew a person who died from COVID-19 (Table 2).

In our sample, 53% (*n* = 220) had never taken an influenza vaccine before, 23% (*n* = 94) indicated that they always take the influenza vaccine when it is available to them, and only 2.6% (*n* = 11) indicated that the influenza vaccine is mandatory in their settings (Table 3).

In general, 14.9% (*n* = 62) were not willing to take the COVID-19 vaccine when it became available to them, 27% (*n* = 114) were hesitant, and 57.8% (*n* = 241) were willing to be vaccinated. Reports in social media influenced the decision to receive the COVID-19 vaccine in 47% (*n* = 195) of our sample, and 31% (*n* = 128) were influenced by celebrities and religious and political leaders when making such a decision (Table 4).

Almost 30% (*n* = 123) of our sample did not trust the government to make the best decision about purchasing the highest quality of vaccine, and 21% (*n* = 88) did not trust pharmaceutical companies to provide credible data on vaccine safety and effectiveness. In addition, 28% (*n* = 117) were not sure that their health centers would have the vaccine available to them when they need it.

When participants were asked about better ways to become immune against the COVID-19, 52% of our sample (*n* = 215) believed that getting sick and acquiring natural immunity is a safer choice than getting the vaccine. This belief was even more emphasized when 33.6% (*n* = 140) were not sure if the benefits of COVID-19 vaccines outweigh their reported side effects and 18.5% (*n* = 77) did not believe they do.

In general, 35.5% (*n* = 148) did not think they have enough information about the COVID-19 vaccines and their safety, and 26.4% (*n* = 110) were not inclined to consent when a new vaccine is introduced, in general. Sixty-six percent of our sample did not hear about anyone who does not want to take the vaccine because of cultural or religious values (Table 5).

In the bivariate analysis, willingness to become vaccinated in this sample was statistically associated with the influence of social media (*H* = 11.97, p = 0.003) and the opinions of celebrities, religious, and political leaders (*H* = 48.89, *p* < 0.001).

Willingness to receive the vaccination was also statistically associated with trust in governments making the right decisions about the vaccine (*H* = 82.32, *p* < 0.001), trust in the pharmaceutical industry to provide credible data about vaccines (*H* = 106.6, *p* < 0.001), and confidence in the health care system to make these vaccines available when needed (*H* = 83.6, *p* < 0.001).

The belief that natural immunity can be a better option to prevent infection (*H* = 8.5, *p* < 0.001) and that the benefits of the vaccine outweigh its risks (*H* = 134.82, *p* < 0.001) were also associated with willingness to receive the vaccine. The attitude towards new vaccines in general (*H* = 143.83, *p* < 0.001) was a predictor of willingness to get the COVID-19 vaccine.

The attitude of the participants in this sample towards introducing new vaccines, in general, was influenced positively by being infected before (*χ^2^* = 8.5, *p* = 0.014) and negatively by knowing someone who died from COVID-19 (*χ^2^* = 5.5, *p* = 0.05).

Additionally, willingness to receive the vaccine was influenced by participants’ position on vaccine hesitancy based on religious or cultural beliefs (*H* = 10.78, *p =* 0.005). Finally, the knowledge about vaccine safety (*H* = 94.5, *p* < 0.001) was also a predictor for willingness to receive the vaccine in a bivariate relationship (Table 6).

Female students were less willing to become vaccinated, U-test = 13,289, *p* < 0.001; on the other hand, the academic year of the participants did not affect their decision. Never getting a flu vaccine (*U* = 13,684, *p* < 0.001) and always getting a flu vaccine (*U* = 8981, *p* < 0.001) significantly influenced willingness to receive the COVID-19 vaccine (Table 7).

In the final model, willingness to become vaccinated was explained by the following factors: Attitudes towards new vaccines (β = 6.23, *p* < 0.001), believing in a favorable risk–benefit ratio (β = 5.64, *p* < 0.001), trust in the pharmaceutical industry (β = 5.92, *p* = 0.001), believing that natural immunity is better than being vaccinated (β = −4.24, *p* < 0.001), and having enough information about the vaccine (β = 4.12, *p* < 0.001). This model explained 46% of the variation in willingness to receive the COVID-19 vaccine (Table 8).

## 4. Discussion

About 58% of dental students in Palestinian educational institutions were willing to get vaccinated against COVID-19 when it becomes available to them. However, willingness to get vaccinated was influenced by attitudes towards new vaccines in general, students’ beliefs about vaccines’ risk–benefit ratio and natural immunity, trust in the pharmaceutical industry, and having enough information about the vaccine.

In the current COVID-19 pandemic, it was expected that HP would be more willing to become vaccinated and encourage the public to do so. Research shows that patients are more likely to accept vaccination when they receive a strong recommendation from their HP [47]. Recent literature reviews on COVID-19 vaccine hesitancy worldwide found that HP hesitancy rates ranged from 27.7% to 78% compared to hesitancy rates in the general public (23.6% to 97%) [48].

Most of the studies that assessed vaccine hesitancy were conducted when vaccines were not yet available, and their mechanisms of action were still unknown [14,49]. At the time of this study, COVID-19 vaccines were already available, and in several countries, the COVID-19 vaccination rollout had already started. In Palestine, the public had been promised the start of vaccination administration at the end of February 2021, but the actual campaign started at the beginning of March 2021 [5]. This study aimed to assess the willingness of dental students to get the COVID-19 vaccine when it becomes available to them. So far, there are no reports from Palestine that investigate the vaccine hesitancy rate in any strata of Palestinian society.

In many countries, dentists were prioritized for vaccination as members of HP teams and because of the well-documented evidence of virus transmission through aerosols and droplets, which places dentists at an even higher risk for potential exposure to the SARS-CoV-2 virus [50]. At the time of writing this current analysis, most dentists were already vaccinated, but dental students are still in the queue. Because of the low availability of the vaccine, the Palestinian Authority’s national strategy for vaccines rollout prioritizes HP at older ages and other community strata such as teachers over young HP students.

This current analysis reported 14.9% rejection and 27% hesitancy towards the COVID-19 vaccine among dental students in this sample. In the United States, dental students showed higher levels of hesitancy (45%) [51]. This can be explained by the timeframe of data collection. The US data were collected in November 2020, when no vaccine was authorized yet by the Food and Drug Administration (FDA) or the WHO while our data were collected at the end of February 2021, when vaccination campaigns including more evidence on their effectiveness and safety were released worldwide.

On comparing Palestine’s data with the global average for dental students’ hesitancy rates collected at the same time using the same methodology, we found numbers in the current analysis to be a bit higher (14.9% vs. 13.9% rejection and 27% vs. 22.5% hesitancy) [38].

Although reports on global and regional trends of hesitancy are very important, specific country data are very valuable to tailor targeted vaccination awareness programs to specific communities. Vaccine hesitancy as documented in the literature is a multifactorial phenomenon that has many cultural and societal influencers. Therefore, to understand the predictors of this behavior, willingness to become vaccinated as a proxy of vaccine hesitancy was further explored in our sample with bivariate and multivariate analysis.

Similar to results of the global survey of dental students, Palestinian students had issues in trusting their government to make the right decisions about the vaccine and the pharmaceutical industry to provide credible data about vaccines, and this mistrust significantly lowered their levels of willingness to receive the vaccination [38].

Female students in this sample were less willing to become vaccinated, and this agrees with the global data on dental students’ hesitancy but disagrees with US dental students’ data as well as dentists’ hesitancy levels in Italy where there was no difference between the two genders in acceptance rate [37,38,51]. A recent meta-analysis on vaccine hesitancy among health workers (HCWs) showed that male participants were more likely to receive the COVID-19 vaccine compared to females when it was available. The previous study suggested that this phenomenon could be because of the reported higher mortality rates among males due to COVID-19 [52].

Interestingly, the year of the study did not affect the willingness rate in this current study. This suggests that advancing through dental training does not better qualify students in sciences related to infectious diseases and vaccines. Almost 36% of the current sample thought that they did not have adequate knowledge about the safety and effectiveness of the COVID-19 vaccines. This lack of information was also a strong predictor of vaccine hesitancy. This is in line with vaccine hesitancy predictors found in other studies for similar populations [37,38].

Unfavorable attitude towards any new vaccine in the current sample was also a predictor of COVID-19 vaccine hesitancy. Being infected by COVID-19 made participants more positive about getting vaccinated in general; however, knowing someone who died because of COVID-19 made participants more negative in their attitude towards vaccination in general. This suggestive border result might reflect how the experience of a relative or a friend death provoked the negative beliefs about disease and vaccination.

The beliefs that natural immunity is a better way to prevent infection and that the benefits of the vaccine do not outweigh its risks were all drivers to the unwillingness to become vaccinated. This also agrees with factors that influenced dentists’ decisions to become vaccinated in another study [37].

Less than half (48%) of the participating students in our sample believed that the COVID-19 vaccine benefits could outweigh their side effects. This suboptimal percentage can be explained by the fact that there was a lack of publicly available safety evidence for the early vaccines used in Palestine, namely Sputnik V and Sinopharm. It is worth mentioning that the passive surveillance systems utilized widely by the drug regulators and governments were found to be inefficient in evaluating the prevalence of COVID-19 side effects, including the common ones like fatigue, headache, and muscle pain; therefore, this study’s findings support the global demand for independent post-marketing studies to evaluate vaccines safety and effectiveness [53,54,55,56].

The influence of social media, opinions of celebrities, and religious and political leaders were also determinants in dental students’ decisions. This was also true in the global survey of dental students’ hesitancy rates [38]. Using public figures and influencers through social media seems to be effective in delivering health promotional messages, as documented in the literature [57]. However, in the current pandemic and with other types of immunizations, social media was used aggressively by the antivaccine campaigns. A study that was conducted in the United States about the uptake of the Human Papilloma Virus (HPV) has shown that individuals’ engagement with anti-vaccine messages on social media has a negative impact on their intentions to get vaccinated [58]. Literature on health communication has shown that emphasizing the benefits of partaking in health behavior rather than portraying the harms of refusing to take the health behavior and focusing on the immediate and personalized benefits rather than distant societal benefits are more effective in delivering health promotion messages [59].

The belief that the decision to get vaccinated should not be based on religious or cultural beliefs positively influenced the willingness to receive the COVID-19 vaccine. Some literature related religiosity with less willingness to receive the vaccine; however, during the current pandemic, many religious institutions encouraged people to become vaccinated, playing the expected role in influencing people’s choices in a positive way [60,61].

Past experience with the influenza vaccine was also a predictor of willingness to get the COVID-19 vaccination. In Palestine, getting the influenza vaccine is optional in all settings, including health care facilities and clinical training programs. Thus, participants who had received the influenza vaccine before exhibited a more positive attitude towards vaccination in general.

This study highlights the need to design a specific curriculum in dental professional training about infectious disease, immunity, and vaccines in addition to public health to enhance knowledge and improve attitudes towards vaccines in general. It also indicates the necessity of building trust between the government, the pharmaceutical industry, and students in healthcare professions. This can be done by arranging meetings and information sessions for students and professionals demonstrating policies related to pandemics and vaccines and speaking openly about the effectiveness and safety of new vaccines.

If dental students, among others in healthcare teams, have positive attitudes towards vaccines, it is hoped that they will share their experiences with patients, family, and friends to encourage vaccine uptake. In the COVID-19 pandemic, many countries are advocating support of dentists administering vaccines to patients to accelerate vaccine rollout during pandemics [51]. Adequate knowledge, trust, and positive attitudes are necessary to make dental students, the future dentists, ready to be part of the healthcare team advocating and recommending receiving vaccines, in general.

Although the current study targeted all dental schools in the occupied Palestinian territories through their FB official social media pages, participants who did not respond to the current survey might have been hesitant to be vaccinated, which can underestimate the true prevalence of vaccine hesitancy among Palestinian dental students. Additionally, the lack of a comparison group of students from different educational backgrounds, such as non-health profession students, limited the ability to assess the influence of the dental professional training on students’ willingness to get the vaccine. However, the progression among academic years from first year to senior year did not influence participants’ decision, which gives an indication that the type of education was not a major player in our current sample responses. Although the use of official students FB groups is very effective to reach students in Palestinian universities, selection bias in the current study can limit the generalizability of our results. Another limitation of the current study design is that the associations among our variables are bidirectional, which makes our statistical model more a descriptive than predictive.

Data collection in the current study was done before vaccination campaigns started, thus first-hand experience with the vaccine was not established yet in these communities. Dental students’ main exposure to information related to the new vaccine was predominantly influenced by media and politics, as these topics had not been included in students’ dental professional training yet.

In general, reporting willingness to receive the vaccine might not translate to real behavior, so we cannot predict whether those who indicated they would receive the vaccine will actually follow through. A future study that follows up on the real behavior of vaccination uptake in this population might validate the current study findings.

## 5. Conclusions

In conclusion, the majority of dental students in Palestinian educational institutions were willing to get vaccinated again COVID-19 when it becomes available to them. However, the negative attitude towards new vaccines, in general, may stem from inadequate information about the favorable risk–benefit ratio of vaccines and the natural immunity vs. the vaccine immunity and the mistrust in pharmaceutical industry transparency, which were all barriers for our sample’ willingness to get vaccinated.

Adequate information about vaccines, their risk–benefit ratios, and natural and acquired immunity are important to build trust and favorable attitudes towards vaccines among dental students. Urgent restructuring of current professional dental training is essential to enable dentists to play the hoped-for role in advocating for and providing vaccinations to their patients, thus speeding up vaccination campaigns among the public as one way to control devastating pandemics like this current one.

## Figures and Tables

**Figure 1 vaccines-09-00954-f001:**
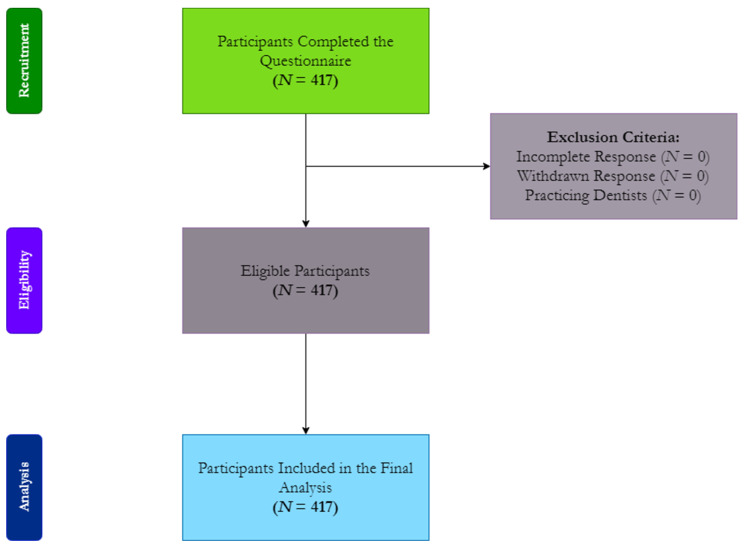
Flow chart of the study’s participants inclusion/exclusion criteria, February—March 2021 (*n* = 417).

**Table 1 vaccines-09-00954-t001:** Demographic characteristics of Palestinian dental students, February–March 2021 (*n* = 417).

Variable	Outcome	Frequency (*n*)	Percentage (*%*)
Gender	Female	295	70.7
	Male	119	28.5
	Prefer not to say	3	0.7
Age Group	≤22 years-old	359	86.1
	>22 years-old	58	13.9
Academic Year	1st Year	31	7.4
	2nd Year	76	18.2
	3rd Year	95	22.8
	4th Year	126	30.2
	5th Year	62	14.9
	Internship	18	4.3
	Fresh Graduate	9	2.2
Clinical Experience	Preclinical	202	48.4
	Clinical	215	51.6

**Table 2 vaccines-09-00954-t002:** COVID-19-related anamnesis of Palestinian dental students, February—March 2021 (*n* = 417).

Variable	Outcome	Frequency (*n*)	Percentage (*%*)
I had been infected by SARS-CoV-2	Yes	81	19.4
	No	336	80.6
I had been caring for someone with COVID-19 infection	Yes	147	35.3
	No	270	64.7
I know someone who had COVID-19 infection	Yes	375	89.9
	No	42	10.1
I personally know someone who had died from COVID-19 infection	Yes	211	50.6
	No	206	49.4

**Table 3 vaccines-09-00954-t003:** Influenza vaccine-related experience of Palestinian dental students, February—March 2021 (*n* = 417).

Variable	Outcome	Frequency (*n*)	Percentage (*%*)
Do you usually take the seasonal influenza vaccine?	Never	220	52.8
	Sometimes	92	22.1
	Always, when I have the chance.	94	22.5
	It is mandatory in my work/study setting.	11	2.6

**Table 4 vaccines-09-00954-t004:** Attitudes of Palestinian dental students towards COVID-19 vaccine, February—March 2021 (*n* = 417).

Variable	Outcome	Frequency (*n*)	Percentage (*%*)
I am willing to take the COVID-19 vaccine once it become available to me	Totally Disagree	36	8.6
	Disagree	26	6.2
	Not Sure	114	27.3
	Agree	83	19.9
	Totally Agree	158	37.9
Do reports you hear/read in the media/on social media make you reconsider the choice to take COVID-19 vaccine?	Yes	195	46.8
	No	110	26.4
	Not Sure	112	26.9
Do celebrities, religious or political leaders influence your decision about getting vaccinated?	Yes	128	30.7
	No	206	49.4
	Not Sure	83	19.9
Do you trust that your government is making decisions in your best interest with respect to what vaccines are provided (e.g., your government purchases the highest quality vaccines available)?	Yes	146	35
	No	123	29.5
	Not Sure	148	35
Do you trust pharmaceutical companies to provide credible data on COVID-19 vaccine safety and effectiveness vaccines?	Yes	200	48
	No	88	21.1
	Not Sure	129	30.9
Do you know anyone who does not take a vaccine because of religious or cultural values?	Yes	93	22.3
	No	274	65.7
	Not Sure	50	12
*If “Yes”, do you agree with these people?*	Yes	21	5
	No	59	14.1
	Not Sure	13	3.2

**Table 5 vaccines-09-00954-t005:** Behaviors of Palestinian dental students towards COVID-19 vaccine, February—March 2021 (*n* = 417).

Variable	Outcome	Frequency (*n*)	Percentage (*%*)
Do you think that there are better ways to prevent COVID-19 than using vaccines (e.g., developing immunity by getting sick and recovered)?	Yes	215	51.6
	No	105	25.2
	Not Sure	97	23.3
Do you feel you have enough information about COVID-19 vaccines and their safety?	Yes	168	40.3
	No	148	35.5
	Not Sure	101	24.2
Do you think that the benefits of COVID-19 vaccines outweigh their reported side effects/adverse reactions?	Yes	200	48
	No	77	18.5
	Not Sure	140	33.6
In general, when a new vaccine is introduced, are you inclined to consent on your vaccination?	Yes	181	43.4
	No	110	26.4
	Not Sure	126	30.2
Do you feel confident that the health center or doctor’s office will have the COVID-19 vaccine you need, when you need them?	Yes	215	51.6
	No	85	20.4
	Not Sure	117	28.1

**Table 6 vaccines-09-00954-t006:** Predictors of Palestinian dental students’ willingness to receive COVID-19 vaccine, February—March 2021 (*n* = 417).

Variable	Outcome	Willingness to Take the COVID-19 Vaccine Once It Is Available			
		Ranks		Test Statistics	
		*n*	Mean Rank	Kruskal-Wallis H	*Sig.*
Do reports you hear/read in the media/on social media make you re-consider the choice to take COVID-19 vaccine?	No	110	180.90	11.971	0.003
	Not Sure	112	203.75		
	Yes	195	227.87		
	Total	417			
Do celebrities, religious or political leaders influence your decision about getting vaccinated?	No	206	172.80	48.894	<0.001
	Not Sure	83	215.04		
	Yes	128	263.34		
	Total	417			
Do you trust that your government is making decisions in your best interest with respect to what vaccines are provided (e.g., your government purchases the highest quality vaccines available)?	No	123	152.70	82.320	<0.001
	Not Sure	148	189.88		
	Yes	146	275.82		
	Total	417			
Do you trust pharmaceutical companies to provide credible data on COVID-19 vaccine safety and effectiveness vaccines?	No	88	121.78	106.609	<0.001
	Not Sure	129	180.48		
	Yes	200	265.78		
	Total	417			
Do you think that there are better ways to prevent diseases than using COVID-19 vaccines (e.g., developing immunity by getting sick and recovered)?	No	105	215.69	8.500	0.014
	Not Sure	97	179.20		
	Yes	215	219.18		
	Total	417			
Do you feel you have enough information about COVID-19 vaccines and their safety?	No	148	154.95	94.498	<0.001
	Not Sure	101	178.47		
	Yes	168	274.98		
	Total	417			
Do you think that the benefits of COVID-19 vaccines outweigh their reported side effects/adverse reactions?	No	77	141.53	134.823	<0.001
	Not Sure	140	148.56		
	Yes	200	277.29		
	Total	417			
In general, when a new vaccine is introduced, are you inclined to consent on your vaccination?	No	110	130.35	143.838	<0.001
	Not Sure	126	169.10		
	Yes	181	284.57		
	Total	417			
Do you feel confident that the health center or doctor’s office will have the COVID-19 vaccine you need, when you need them?	No	85	141.44	83.603	<0.001
	Not Sure	117	167.45		
	Yes	215	258.32		
	Total	417			
Do you know anyone who does not take a vaccine because of religious or cultural values?	No	274	206.69	3.272	0.195
	Not Sure	50	190.81		
	Yes	93	225.58		
	Total	417			
*If “Yes”, do you agree with these people?*	No	59	53.60	10.780	0.005
	Not Sure	13	34.58		
	Yes	21	36.14		
	Total	93			

**Table 7 vaccines-09-00954-t007:** Predictors of Palestinian dental students’ willingness to receive COVID-19 Vaccine, February—March 2021 (*n* = 417).

Variable	Outcome	Willingness to Take the COVID-19 Vaccine Once It Is Available			
		Ranks		Test Statistics	
		*N*	Mean Rank	Mann-Whitney U	*Sig.*
Gender	Female	295	193.05	13,289.000	<0.001
	Male	119	243.33		
	Total	414			
Clinical Experience	Preclinical	202	211.98	21,113.500	0.609
	Clinical	215	206.20		
	Total	417			
I “never” took the seasonal influenza vaccine	No	197	249.54	13,684.000	<0.001
	Yes	220	172.70		
	Total	417			
I “sometimes” take the seasonal influenza vaccine	No	325	205.55	13,830.000	0.252
	Yes	92	221.17		
	Total	417			
I “always” take the seasonal influenza vaccine	No	323	189.81	8981.500	<0.001
	Yes	94	274.95		
	Total	417			
It is “mandatory” take the seasonal influenza vaccine in my work/study setting	No	406	207.36	1566.500	0.078
	Yes	11	269.59		
	Total	417	193.05		

**Table 8 vaccines-09-00954-t008:** Model of Palestinian dental students’ attitudes towards COVID-19 vaccine, February—March 2021 (*n* = 417).

	Standardized Coefficients		
Model	*Beta*	*t*	*Sig.*
(Constant)		9.736	<0.001
In general, when a new vaccine is introduced, are you inclined to consent on your vaccination?	0.479	6.426	<0.001
If “Yes”, do you agree with these people?	−0.333	−4.450	<0.001
Never	−0.287	−3.751	<0.001
Do you trust pharmaceutical companies to provide credible data on COVID-19 vaccine safety and effectiveness vaccines?	0.190	2.519	0.014
Do you feel you have enough information about COVID-19 vaccines and their safety?	0.248	3.219	0.002
Do you trust that your government is making decisions in your best interest with respect to what vaccines are provided (e.g., your government purchases the highest quality vaccines available)?	−0.182	−2.267	0.026
Do you think that there are better ways to prevent diseases than using COVID-19 vaccines (e.g., developing immunity by getting sick and recovered)?	−0.175	−2.193	0.031

## Data Availability

The data that support the findings of this study are available from the corresponding author upon reasonable request.

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
