# Peer review of "Predictors of Willingness to Receive COVID-19 Vaccine: Cross-Sectional Study of Palestinian Dental Students"

_vaccines, 2021, doi:10.3390/vaccines9090954_

Round 1
Reviewer 1 Report
Thank you for the opportunity to review this paper. Overall, the paper was well written and argued. The comments below are more about trying to bring the argument out more and other comments are about developing the work more broadly.
- More discussion on the vaccine roll out would be useful - especially outlining what were the barriers and challenges of this vaccine roll out.
- Given the ‘success’ the COVID vaccination in Israel- are there any issues from the Arab Israeli community that might be useful for the study to acknowledge?
- From line 72- more context would be good when examining the context of vaccine hesitancy, but this I mean links to the lack of trust in science literature or expertise eg Tom Nichols etc
- In terms of methodology- I feel the dental student sample is somewhat limited and may lack generalisability - I would encourage the author for future studies to try a more comparative approach with other groups or countries as well.
- Comment about natural immunity line 214 needs more context- this sentence is a bit rough and could be used out of context. I would add a little more detail and context to this.
- The section on the influence of social media line 330, seems a bit light- are you able to give more detail what forms of social media and why there were important?
- You might want to include in the study some sentence on how the background of students have impacted the study eg the influence of social class etc.
Author Response
Dear Reviewer
We are delighted to have the opportunity to revise and resubmit our manuscript titled “Predictors of Willingness to Receive COVID-19 Vaccine: Cross-sectional Study of Palestinian Dental Students" (Manuscript ID vaccines-1326909).
We have considered all remarks provided by all of the reviewers. Please find appended a revised version of the manuscript (with track changes highlighted) and a point-by-point rebuttal to all comments raised as detailed below. We hope our responses are satisfactory in addressing the criticisms and suggestions.
We hope the revised manuscript will be in an acceptable format. Thank you for your kind contribution.
- More discussion on the vaccine rollout would be useful - especially outlining what were the barriers and challenges of this vaccine rollout.
Answer: Done. Thank you
- Given the ‘success’ the COVID vaccination in Israel- are there any issues from the Arab Israeli community that might be useful for the study to acknowledge?
Answer: Thank you for your comment. Our sample is limited to dental students in Palestinian universities. Many of the Palestinians who live behind the green line aim Palestinian universities to study dentistry; thus they are kind of represented in this sample. However, we asked a hypothetical question about willingness of getting vaccine not the actual status and the location of residence was not statistically different.
- From line 72- more context would be good when examining the context of vaccine hesitancy, but this I mean links to the lack of trust in science literature or expertise eg Tom Nichols etc
Answer: Thank you for your comment. I added more context to this paragraph as suggested.
- In terms of methodology- I feel the dental student sample is somewhat limited and may lack generalizability - I would encourage the author for future studies to try a more comparative approach with other groups or countries as well.
Answer: Thank you for your comment. This data is part of a global study that targeted dental students as an example of future healthcare providers and was published in Vaccine. However, we believe that vaccine hesitancy is influence by cultural, religious and societal norms, therefore, we thought studying this phenomenon at a country level is important and help that country and maybe neighbouring one to understand this phenomenon in their own context. This was illustrated in the highlighted text in lines 74-82. We will in the future try to extend this study to include more professions and/or the general population.
- Comment about natural immunity line 214 needs more context- this sentence is a bit rough and could be used out of context. I would add a little more detail and context to this.
Answer: Thank you for your comment. I rephrased the whole paragraph to sound clearer.
- The section on the influence of social media line 330, seems a bit light- are you able to give more detail what forms of social media and why there were important?
Answer: Thank you for your comment. I added more details and two references to this paragraph.
- You might want to include in the study some sentence on how the background of students have impacted the study eg the influence of social class etc.
Answer: Thank you for your comment. I added more about this in the limitations section
Sincerely,
Reviewer 2 Report
In the manuscript entitled: “Predictors of Willingness to Receive COVID-19 Vaccine among Dental Students”, the authors assessed the predictors related to the willingness of Palestinian dental students to receive the COVID-19 vaccine when it becomes available
The authors found that 57.8% were willing to take the COVID-19 vaccine when it became available to them, 27% were hesitant, and 14.9% were not willing to get vaccinated. The final regression model explained 46% of the variation in the willingness to receive the COVID-29 19 vaccine as follows: attitudes towards new vaccines (p<0.001), believing in a favorable risk-benefit ratio (p<0.001), trust in the pharmaceutical industry (p=0.001), believing that natural immunity is better than being vaccinated (β=-4.24, p<0.001) and having enough information about the vaccine (p<0.001).
The authors concluded that adequate information about vaccines, their risk-benefit ratios, and natural and acquired immunity are important to build trust and favorable attitudes towards vaccines among future dentists.
Major comments:
In general, the idea and innovation of this study, regards analysis of Willingness to Receive COVID-19 Vaccine among Dental Students is quite interesting, because the role of these factors in medicine are still validated but further studies on this topic could be an innovative issue in this field could be not open a creative matter of debate in literature by adding new information. Moreover, there are few reports in the literature that studied this interesting topic with this kind of study design.
The study was well conducted by the authors; However, there are some concerns to revise that are described below.
The introduction section not well resumes the existing knowledge regarding the important factor linked with importance of COVID vaccine in dental field.
However, as the importance of the topic, the reviewer strongly recommends, before a further re-evaluation of the manuscript, to update the literature through read, discuss and must cites in the references with great attention all of those recent interesting articles, that helps the authors to better introduce and discuss the role of vaccination in dental students.
In the material and methods section, should better clarify the overall impact on quality of life in dental students
The discussion section appears not well organized with the relevant paper that support the conclusions, even if the authors should better discuss the relationship between covid infections and reduced daily work. The conclusion should reinforce in light of the discussions.
In conclusion, I am sure that the authors are fine clinicians who achieve very nice results with their adopted protocol. However, this study, in my view does not in its current form satisfy a very high scientific requirement for publication in this journal.
Minor Comments:
Abstract:
- Better formulate the abstract section by better describing the aim of the study
Introduction:
- Please refer to major comments
Discussion
- Please add a specific sentence that clarifies the results obtained in the first part of the discussion
- Page 10 last paragraph: Please reorganize this paragraph that is not clear
Author Response
Dear Reviewer
We are delighted to have the opportunity to revise and resubmit our manuscript titled “Predictors of Willingness to Receive COVID-19 Vaccine: Cross-sectional Study of Palestinian Dental Students" (Manuscript ID vaccines-1326909).
We have considered all remarks provided by all of the reviewers. Please find appended a revised version of the manuscript (with track changes highlighted) and a point-by-point rebuttal to all comments raised as detailed below. We hope our responses are satisfactory in addressing the criticisms and suggestions.
We hope the revised manuscript will be in an acceptable format. Thank you for your kind contribution.
Major comments:
- In general, the idea and innovation of this study, regards analysis of Willingness to Receive COVID-19 Vaccine among Dental Students is quite interesting, because the role of these factors in medicine are still validated but further studies on this topic could be an innovative issue in this field could be not open a creative matter of debate in literature by adding new information. Moreover, there are few reports in the literature that studied this interesting topic with this kind of study design.
Answer: Thank you for this encouraging feedback.
- The study was well conducted by the authors; However, there are some concerns to revise that are described below.
Answer: Thank you for your time revising this article.
- The introduction section not well resumes the existing knowledge regarding the important factor linked with importance of COVID vaccine in dental field. However, as the importance of the topic, the reviewer strongly recommends, before a further re-evaluation of the manuscript, to update the literature through read, discuss and must cites in the references with great attention all of those recent interesting articles, that helps the authors to better introduce and discuss the role of vaccination in dental students.
Answer: The introduction section was revised and new text and references were added. Thank you!
- In the material and methods section, should better clarify the overall impact on quality of life in dental students
Answer: We have referred to few publications done by the team that assessed aspects of the impact of the pandemic on dental students in general. I added one study about psychological distress among dental students in Palestine.
- The discussion section appears not well organized with the relevant paper that support the conclusions, even if the authors should better discuss the relationship between covid infections and reduced daily work. The conclusion should reinforce in light of the discussions.
Answer: I added some text to the discussion and some references to clarify some unclear points. I hope it reads better now. Thank you!
- In conclusion, I am sure that the authors are fine clinicians who achieve very nice results with their adopted protocol. However, this study, in my view does not in its current form satisfy a very high scientific requirement for publication in this journal.
Answer: I hope the modification we did improved the quality of the paper. Thank you for your insightful comments!
Abstract:
- Better formulate the abstract section by better describing the aim of the study
Answer: I modified the aim in the last paragraph of the introduction. Thank you!
Discussion
- Please add a specific sentence that clarifies the results obtained in the first part of the discussion
Answer: Done. Thank you!
- Page 10 last paragraph: Please reorganize this paragraph that is not clear
Answer: Done. Thank you!
Sincerely,
Reviewer 3 Report
Title should specify study design and location
Keywords please check and use Mesh term when applicable
Methods: please add inclusion exclusion criteria
What was the assumption behind the estimation of the sample size? Information provided is not enought.
Please, add the questionnaire as supplementary material
many information regarding the setting is reported throughout the manuscript. I suggest to combine all of them in one section titled setting in the methods section.
What was the denominator based on which responsa rate was estimate? Authors spoke about more than 3500 students, and it seems not in agreement with the 41.7% of the response rate.
Results in lines 236-237 are, in my view and knowledge, in contrast with previous evidence. This aspect was only marginally discussed in the discussion section lines 312-316. Please, make some more clarification and more comparison with previous research.
line 244, please take in mind that personal considerations are not allowed in the results section. So I would suggest to remove the term obviously. Moreover, if authors considered it obvious, it should be discussed in the discussion section. It is completely missing. Please add. I suggest adding consideration also comparing this result with previous research.
Discussion should be restructured. Usually, the discussion should be maintain the following structure:
- Summay of may results (main messages authors want to bring out to the readers). this part is completely missing
- internal validity: internal comparison of the results found within the study
- external validity: comparison of own results with previous evidence
- strengths and limitations: authors should spoke about potential biases and how they tried to solve them.
In my view discussion should be deeply revised.
in the limitations, authors did not mention facebook as a potential limitation. However, it should be considered because fake profile migh exist, it could not be used by all subjects and bubble chain effect could be there. Please comment on it. Moreover, more consideration should be posed on the selection bias. Actually, authors do no have any information on non-responders. A statement on this aspect should be added.
The unexpected results could be due to potential selection bias. Please make a consideration.
The authors declared no conflict of interest, however, it seems that some authors have the same affiliation of the institute that granted the study. Please clarify and add the role of the funder.
coclusions should be mainly based on results obtained, instead of speculations
Extensive english revision is needed. Just as examples (Fresh graduated is not correct in english. make sure you are speaking about 22 countries or county, as reported in line 116
Author Response
Dear Reviewer
We are delighted to have the opportunity to revise and resubmit our manuscript titled “Predictors of Willingness to Receive COVID-19 Vaccine: Cross-sectional Study of Palestinian Dental Students" (Manuscript ID vaccines-1326909).
We have considered all remarks provided by all of the reviewers. Please find appended a revised version of the manuscript (with track changes highlighted) and a point-by-point rebuttal to all comments raised as detailed below. We hope our responses are satisfactory in addressing the criticisms and suggestions.
We hope the revised manuscript will be in an acceptable format. Thank you for your kind contribution.
- Title should specify study design and location
Answer: Done. Thank you
- Keywords please check and use Mesh term when applicable
Answer: Checked. Thank you
- Methods: please add inclusion exclusion criteria
Answer: Done. Thank you
- What was the assumption behind the estimation of the sample size? Information provided is not enough.
Answer: The total population, the margin of error and the level of confidence were provided. What other pieces of info do you suggest? Thank you
- Please, add the questionnaire as supplementary material
Answer: Added. Thank you for the recommendation!
- What was the denominator based on which responsa rate was estimate? Authors spoke about more than 3500 students, and it seems not in agreement with the 41.7% of the response rate.
Answer: This was a wrong estimation. I removed this part. Thank you for drawing my attention to this.
- Results in lines 236-237 are, in my view and knowledge, in contrast with previous evidence. This aspect was only marginally discussed in the discussion section lines 312-316. Please, make some more clarification and more comparison with previous research.
Answer: Yes, this was an interesting result. Unfortunately, I searched the literature for results relevant to this point and couldn’t find any. Some news reports from the United States demonstrated the death in the families did not push people to get vaccinated.
- line 244, please take in mind that personal considerations are not allowed in the results section. So I would suggest to remove the term obviously. Moreover, if authors considered it obvious, it should be discussed in the discussion section. It is completely missing. Please add. I suggest adding consideration also comparing this result with previous research.
Answer: Done. Thank you for the suggestion, it improved this paragraph a lot.
- Discussion should be restructured. Usually, the discussion should be maintain the following structure:
Answer: I rewrote parts of the discussion and added some text and references to flow more easily
- in the limitations, authors did not mention facebook as a potential limitation. However, it should be considered because fake profile migh exist, it could not be used by all subjects and bubble chain effect could be there. Please comment on it. Moreover, more consideration should be posed on the selection bias. Actually, authors do no have any information on non-responders. A statement on this aspect should be added.
Answer: I added this to the limitation section. Thank you
- The unexpected results could be due to potential selection bias. Please make a consideration.
Answer: Added this to the limitations section. Thank you
- The authors declared no conflict of interest, however, it seems that some authors have the same affiliation of the institute that granted the study. Please clarify and add the role of the funder.
Answer: We would like to thank the reviewer for giving us the opportunity to clarify this point. In the "Funding" section, the projects which were acknowledged are either funded directly by Masaryk University or by the Czech government through Masaryk University. The authors (A.R., A.P., J.K., and M.K.) belong to Masaryk University and their work is supported by these projects, especially for supporting open-access publishing. Therefore, we did not believe that any of us have a financial conflict of interest, as it's normal in academia to be supported by our own institution.
- coclusions should be mainly based on results obtained, instead of speculations
Answer: The conclusion was revised and recommendations were checked.
- Extensive english revision is needed. Just as examples (Fresh graduated is not correct in english. make sure you are speaking about 22 countries or county, as reported in line 116
Answer: Corrected. Thank you
Sincerely,
Reviewer 4 Report
The study was to assess the predictors related to the willingness of dental students to receive the COVID-19 vaccine. The study was a cross-sectional study among the sample of dental students in the Palestinian territories. The below factors were included in the study for Willingness to get the COVID-19 vaccine.
Demographic characteristics, COVID-19-related experiences, beliefs and knowledge about the vaccine, attitudes, and other factors defined by the WHO SAGE Vaccination Hesitancy Questionnaire.
The study result noted that 57.8% (n=241) were willing to take the COVID-19 vaccine when it became available to them, 27% (n=114) were hesitant, and 14.9% did not interested to get vaccination. The study utilize the data of regression model, which explained the variation in the willingness to receive the COVID- 19 vaccines.
These factors were attitudes towards new vaccines, believing in a favorable risk-benefit ratio, faith in the pharmaceutical industry, believing that natural immunity is better than being vaccinated.
Below are the comments to be incorporated into the manuscript
- Line 47-48 It is more to explore and discuss the reference
- Line 62-63 These are general lines and the authors put references here, I think no need to add references here
- Line 70-71 reference 14 Please discuss the paper comprehensibly
- Line 88-89 Please discuss these papers here
- Section 2.2 The representation of recruited participants need a flow chart that could highlight the exclusion or inclusion of the participants
- Section 2.4 The studied subjects were from Palestine and ethical consideration was approved from a different country
- Line 199-201 what does it indicate?
- They are medical practitioners and they did not trust on pharmaceutical company? if they do not have positive perceptions how come the general population follow
- Line 220 please follow one representation either numerical or words (Sixty-six percentage)
- is this was the explanation of table 8?
- The first para of discussion need to highlight the result of this paper, please add up the finding here
- Line 264 Many? The authors have to mention more references
- Line 269-271 No need to repeat
- Line 273-274 Do not start with many countries as it was difficult to cite and better to highlight one study
- Line 315-316 The lines need to modify in a better way
- Line 324-329 Long sentence break into 2 sentences
- Need to share the limitations of the study
- The references need to be changed as only one author used EXCESSIVELY Riad et al in the manuscript.
Author Response
Dear Reviewer
We are delighted to have the opportunity to revise and resubmit our manuscript titled “Predictors of Willingness to Receive COVID-19 Vaccine: Cross-sectional Study of Palestinian Dental Students" (Manuscript ID vaccines-1326909).
We have considered all remarks provided by all of the reviewers. Please find appended a revised version of the manuscript (with track changes highlighted) and a point-by-point rebuttal to all comments raised as detailed below. We hope our responses are satisfactory in addressing the criticisms and suggestions.
We hope the revised manuscript will be in an acceptable format. Thank you for your kind contribution.
- Line 47-48 It is more to explore and discuss the reference
Answer: Thank you for your comment. I added more context to this point through the whole introduction. New text is highlighted.
- Line 62-63 These are general lines and the authors put references here, I think no need to add references here
Answer: Thank you for your comments. I added some text that needs references now.
- Line 70-71 reference 14 Please discuss the paper comprehensibly
Answer: Thank you for your comment. I added more info from this paper to the text.
- Line 88-89 Please discuss these papers here
Answer: Done. Thank you!
- Section 2.2 The representation of recruited participants need a flow chart that could highlight the exclusion or inclusion of the participants
Answer: Added (Figure 1). Thank you!
- Section 2.4 The studied subjects were from Palestine and ethical consideration was approved from a different country
Answer: This data was extracted from a global survey that was conducted by Masaryk University. Administrative approval to conduct the study was granted. This line was added to the text. Thank you!
- They are medical practitioners and they did not trust on pharmaceutical company? if they do not have positive perceptions how come the general population follow
Answer: They are still dental students and as the results indicate, progress in years of study did not influence their decision to get vaccinated. That’s why we suggested more focus on vaccinology, public health and immunology in the curriculum.
- Line 220 please follow one representation either numerical or words (Sixty-six percentage)
Answer: We wrote the numbers in words only when they come in the beginning of a sentence. Thank you!
- The first para of discussion need to highlight the result of this paper, please add up the finding here
Answer: Done. Thank you!
- Line 264 Many? The authors have to mention more references
Answer: Done. Thank you!
- Line 273-274 Do not start with many countries as it was difficult to cite and better to highlight one study
Answer: A reference was added. Thank you!
- Need to share the limitations of the study
Answer: Limitations were explored more. Thank you!
- The references need to be changed as only one author used EXCESSIVELY Riad et al in the manuscript.
Answer: Thank you for the recommendation. We decreased the references of our previous work.
Sincerely,
Reviewer 5 Report
- this questionnaire and its report is not ‘novel’, so that I assess the priority as (very) low. Furthermore, the response rate (41.7%) is also quite low, so that the reliability that the results can be applied to the full cohort or other similar cohorts of ‘students involved in health care’ is further limited. Finally, related to the statistics, I disagree that this is a prediction model, but rather a descriptive model.
Additional reflections
The vaccines effectiveness and safety profile are not dependent on the ‘parts of the world’, so that this sentence (line 68-70) should be reconsidered. It is rather how different populations assess the effectiveness and safety profile that differs.
Are there any differences in characteristics between participants and non-participants.
Reference 41: why are the findings not pooled in an international initiative ?
Author Response
Dear Reviewer
We are delighted to have the opportunity to revise and resubmit our manuscript titled “Predictors of Willingness to Receive COVID-19 Vaccine: Cross-sectional Study of Palestinian Dental Students" (Manuscript ID vaccines-1326909).
We have considered all remarks provided by all of the reviewers. Please find appended a revised version of the manuscript (with track changes highlighted) and a point-by-point rebuttal to all comments raised as detailed below. We hope our responses are satisfactory in addressing the criticisms and suggestions.
We hope the revised manuscript will be in an acceptable format. Thank you for your kind contribution.
- The response rate (41.7%) is also quite low, so that the reliability that the results can be applied to the full cohort or other similar cohorts of ‘students involved in health care’ is further limited.
Answer: Thank you for your comments. I really understand your concerns. However, surveys among health professionals usually do not exceed 30% response rate. We hope by obtaining enough sample size based on sample size calculations and representing all dental schools in Palestinian territories that we reached a reasonable representative sample.
- Finally, related to the statistics, I disagree that this is a prediction model, but rather a descriptive model.
Answer: Thank you. I agree, I added this to the limitations.
- The vaccines effectiveness and safety profile are not dependent on the ‘parts of the world’, so that this sentence (line 68-70) should be reconsidered. It is rather how different populations assess the effectiveness and safety profile that differs.
Answer: Thank you for indicating this. I meant “Vaccine hesitancy”. The text was clarified.
- Are there any differences in characteristics between participants and non-participants.
Answer: We were not able to assess this. However, both genders, all academic years and all dental schools were represented in our sample.
- Reference 41: why are the findings not pooled in an international initiative?
Answer: It was merely a descriptive study. Future analysis will pool the data to study factors related to vaccine hesitancy in more details.
Sincerely,
Round 2
Reviewer 2 Report
The authors have well addressed to all comments raised by the reviewer. There are no further issues in the present version of the manuscript.
Reviewer 3 Report
I'm satisfied with the changes provided
Reviewer 4 Report
The authors have addressed all concerns.
Reviewer 5 Report
nothing to add
This manuscript is a resubmission of an earlier submission. The following is a list of the peer review reports and author responses from that submission.